# Object Detection Using Multi-Scale Balanced Sampling

**Hang Yu *** 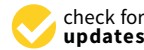**, Jiulu Gong and Derong Chen**

School of Mechatronical Engineering, Beijing Institute of Technology, Beijing 100811, China;
lujiugong@bit.edu.cn (J.G.); cdr@bit.edu.cn (D.C.)
* Correspondence: yuh_0111@bit.edu.cn; Tel.: +86-010-88521997

**Abstract:** Detecting small objects and objects with large scale variants are always challenging for deep learning based object detection approaches. Many efforts have been made to solve these problems such as adopting more effective network structures, image features, loss functions, etc. However, for both small objects detection and detecting objects with various scale in single image, the first thing should be solve is the matching mechanism between anchor boxes and ground-truths. In this paper, an approach based on multi-scale balanced sampling(MB-RPN) is proposed for the difficult matching of small objects and detecting multi-scale objects. According to the scale of the anchor boxes, different positive and negative sample IOU discriminate thresholds are adopted to improve the probability of matching the small object area with the anchor boxes so that more small object samples are included in the training process. Moreover, the balanced sampling method is proposed for the collected samples, the samples are further divided and uniform sampling to ensure the diversity of samples in training process. Several datasets are adopted to evaluate the MB-RPN, the experimental results show that compare with the similar approach, MB-RPN improves detection performances effectively.

**Keywords:** object detection; small object; multi-scale sampling; balanced sampling

## 1. Introduction

Object detection is a kind of approaches for objects localization and category classification in digital images, which is one of the most challenging branches in the field of computer vision.

The early approaches are based on handcrafted image features. In [1], a pedestrian detection system is proposed with histogram of oriented gradients(HOG) feature and support vector machine(SVM). In [2], deformable part-based model(DPM) is proposed which enhanced detection accuracy by utilizing HOG features of the whole and part of objects. As the peak of handcraft feature based detection approach, the detection performances of DPM are still not ideal, due to the lack of effective representation of features. Besides, since hand craft feature extractor are always designed for specific object types, hence often result in low robustness in dealing with different category of objects.

In recent years, several object detection approaches based on convolutional neural networks (CNN) are proposed [3–5]. Image features are supervise trained by measuring error between prediction and annotated ground-truth in large-scale object detection datasets. Compare with handcraft features the CNN features' representation ability and robustness against various types of objects are both significantly enhanced. Therefore the detection performances are highly improved. Although deep learning object detection approaches have shown state of the art performance for general object detection, they are still limited in detecting small objects and the performances in detection various scale objects in single input image is also not ideal. The reasons for a low detection performances are as the follow:

1. The proportion of small objects in the image are always relative low, which means they might be excluded from the training process due to improper network hyperparameter settings. However, small objects often have low resolutions and less image information which means the difficulty of training small objects are always higher than general objects, the lack of sampling will further result in a low quality features extraction by the deep neural networks.
2. In natural scenes, the scale of objects are distributed stochastic, which means within a single image there might be objects with large scale variants, it is easy to take the majority of samples and ignore other objects in the process of training.

Overall, in order to further improve detection performances especially in detecting objects mentioned above, the number of small objects and proportion of various scale object in training samples are both important.In this paper an end-to-end object detection approach with multi-scale balanced sampling is proposed to improve the matching mechanism and ensure scale diversity in training samples. The key contribution of the approach is summarized as follow:

1. The samples' matching conditions is adjusted according to the objects' scale so that the the small objects are easier to be matched, which enhance the training samples of small objects.
2. The sample set is divided into multiple intervals according to the samples' scale and their corresponding discrimination difficulty. In addition, each interval is sampled in a balanced fashion to preserve the diversity of sample types during the classification and positioning network training process and to further ensure that the algorithm does not tend to detect a specific type of samples while ignoring the others.
3. Evaluation between proposed approach and others on several benchmarks is proposed, the experiment results show that the detection performances are better than other similar approaches.

The remainder of this paper is organized as follows. In Section 2, background and related works are introduced. In Section 3, framework and implement detail of proposed approach are introduced. Section 4 presents the experiment results and comparisons with other similar approaches. Finally, Section 5 conclude the proposed approach.

## 2. Related Works

In [6], a deep neural network based object detection approach called RCNN is first introduced. It is composed with three parts: First, by adopting selective search algorithm, RCNN generate a series of candidate region, each of them may responsible for detecting a specific object in the image. Second, extract feature of candidate regions by CNN, the network will be train supervised by measuring the error between prediction and ground-truths. Finally, SVM and bounding box regression is adopted to finetune the predicted results. The framework of RCNN is similar to tradition approaches except CNN is adopt to extract image features instead of handcraft features. In [7], an approach called Fast RCNN is proposed, it take the whole image as inputs, then crop the corresponding features of each candidate region and map them to a uniform size by region of interest(ROI) pooling, finally feed the mapped features into classification and regression network to acquire its category and localization. The fast RNN integrate coarse and finetune process in RCNN which improve both the detection performances and efficiency. In [5], an end-to-end CNN based object detection framework called Faster RCNN is introduced. Instead of generate candidate by selective search, this approach proposed region proposal network(RPN), it first generate a series of anchor boxes with different scale and aspect ratio, each of them is responsible for detecting object or not is depend on the inter section of union(IOU) between its coordinate and annotated ground-truth. As the first end-to-end object detection approach, Faster RCNN laid the foundation of subsequent deep neural network based object detection approaches. In [3], an approach which integrate the function of RPN and classification/regression network in one series convolution layers called SSD is introduced. At present, approaches whose framework are similar to Faster RCNN are summarized as two-stage approaches, in contrast the approaches like SSD are called one-stage approaches.

Based on backbones such as Faster RCNN and SSD, variety of approaches were proposed to enhance the detection performances. For example: Feature Pyramid Networks(FPN) proposed an pyramid architecture image feature extractor, objects in different resolution are arranged to corresponding layers, compare with the origin Faster RCNN, in dealing with a specific object FPN will provide more proper feature [8]. Cascade RCNN proposed training process that discrimination IOU threshold is gradually increased, which makes the classification and regression network training in a easy-to-hard way [9]. RetinaNet proposed focal loss which could enlarge the weight of hard samples, which makes the training process focuses on the hard samples [10]. Libra RCNN proposed a balanced sampling method, feature extraction and loss function in training process [11]. SRetinaNet propose an anchor optimization method which will help detecting small objects with specific parameter setting [12]. GA-RPN propose an anchor optimization method by combining anchor box with semantic features [13].

Regardless approaches being one or two stage, the fundament for object detection is the matching mechanism of anchor and ground-truth, which determines how many samples can be included in the network training. Therefore, it is important to propose a proper matching mechanism for enhancing the detection performances. However, the stochastic of objects scale poses challenges to the matching mechanism [12,13].

## 3. Proposed Method

### 3.1. Framework Overview

The overview of proposed approach is shown in Figure 1, where the green cubes denote image feature extracting process, the pink cube denote MB-RPN module and purple cube denote classification/regression networks for finetune. Table 1 shows the details of the network. The specific steps are as follows:

1. Take an digital image as inputs, feed it into ResNet-101 pre-trained network so that the image features are extracted from shallow to deep using Conv1~Conv5 [14].
2. By adopting MB-RPN, samples are dynamically selected according to scale and proportion. MB-RPN can be further decomposed into two parts: multi-scale and balanced sampling.
3. The MB-RPN calculation results are then transformed to the same size through ROI Pooling.
4. The candidate box is then sent to the classification and regression network to obtain the object category and its location.

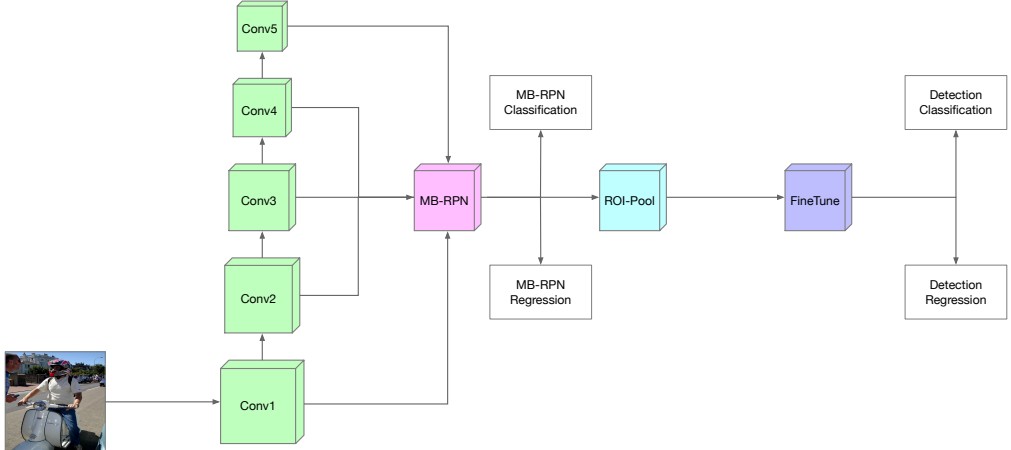

**Figure 1.** Architecture of Proposed Method.

**Table 1.** Details of Network.

| | Input | Shape | Layer | Output |
|---|---|---|---|---|
| | Image | $600 \times 600 \times 3$ | $conv\_1 * 1, stride\, 2, 64$ | Conv1 |
| | Conv1 | $300 \times 300 \times 64$ | $conv\_3 * 3, maxpool, stride\, 2, 64$ | Pool1 |
| ResNet-101 | Pool1 | $150 \times 150 \times 64$ | $\begin{bmatrix} conv\_1 * 1, 64 \\ conv\_3 * 3, 64 \\ conv\_1 * 1, 256 \end{bmatrix} \times 3$ | Conv2 |
| | Conv2 | $150 \times 150 \times 256$ | $\begin{bmatrix} conv\_1 * 1, 128 \\ conv\_3 * 3, 128 \\ conv\_1 * 1, 512 \end{bmatrix} \times 4$ | Conv3 |
| | Conv3 | $75 \times 75 \times 512$ | $\begin{bmatrix} conv\_1 * 1, 256 \\ conv\_3 * 3, 256 \\ conv\_1 * 1, 1024 \end{bmatrix} \times 3$ | Conv4 |
| | Conv4 | $38 \times 38 \times 1024$ | $\begin{bmatrix} conv\_1 * 1, 512 \\ conv\_3 * 3, 512 \\ conv\_1 * 1, 2945 \end{bmatrix} \times 3$ | Conv5 |
| MB-RPN | Pool1 | $150 \times 150 \times 64$ | $conv\_3 * 3, 256$ | $MB - RPN_1$ |
| | Conv2 | $75 \times 75 \times 256$ | $conv\_3 * 3, 256$ | $MB - RPN_2$ |
| | Conv3 | $38 \times 38 \times 512$ | $conv\_3 * 3, 256$ | $MB - RPN_3$ |
| | Conv4 | $19 \times 19 \times 1024$ | $conv\_3 * 3, 256$ | $MB - RPN_4$ |
| | Conv5 | $10 \times 10 \times 2048$ | $conv\_3 * 3, 256$ | $MB - RPN_5$ |
| FineTune | Pool1 | $14 \times 14 \times 128$ | $conv\_3 * 3, 256$ | $FineTune_1$ |
| | Conv2 | $28 \times 28 \times 256$ | $conv\_3 * 3, 256$ | $FineTune_2$ |
| | Conv3 | $56 \times 56 \times 374$ | $conv\_3 * 3, 256$ | $FineTune_3$ |
| | Conv4 | $112 \times 112 \times 512$ | $conv\_3 * 3, 256$ | $FineTune_4$ |
| | Conv5 | $224 \times 224 \times 640$ | $conv\_3 * 3, 256$ | $FineTune_5$ |

## *3.2. Multi-Scale Sample Discrimination*

The main factors affecting the sample matching in training process are the scale of the anchor box and the labelling result IOU discrimination threshold. The large difference between the scale of the small object and the anchor box makes match difficult under the existing discrimination conditions.

For the sampling process of the positive samples, Figure 2a shows the matching results when default shape of FPN anchor boxes is adopted and the discrimination threshold of IOU> 0.7, where the red rectangle indicates the manually marked area containing the object. As it is seen, none of the anchor boxes can be successfully matched with the object area. Therefore, the image is unable to guide the network parameter training because it does not contain any positive sample during the the training process. Figure 2b shows the matching results for the case where the scale of the anchor box is reduced by half. The green rectangle indicates the labeled samples in the training set, and the red rectangle indicates the corresponding matching anchor box, the sample matching results are still far from ideal.

For small objects, the default IOU threshold of FPN is a stringent condition, resulting in a poor matching even in the case where the anchor box scale is reduced. Also, the design of the anchor boxes should fully consider the objects in the image data set with different sizes. Therefore, simple reduction of the anchor box scale might, in return, result in matching failure for the object samples with a normal size. Hence, it is hard to improve the object matching probability solely by reducing the scale of the anchor box for detecting small objects.

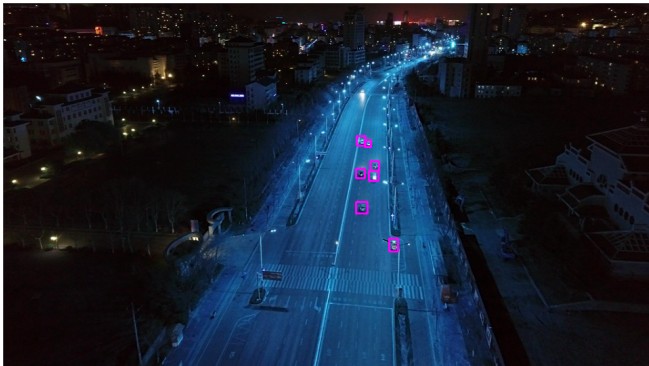

(**a**)default FPN anchor sizes

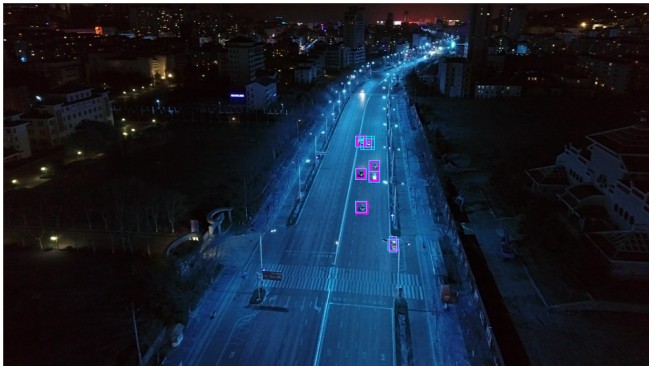

(**b**)half of the default FPN anchor sizes

**Figure 2.** Sampling Results Example on Small objects.

To address the above issue, multi-scale positive sampling approach with dynamic IOU discrimination threshold is proposed. The FPN method has designed five scale-level anchor boxes, namely, A1∼A5 according to different scale sizes. According to the scale of anchor boxes, the approach divide three different positive sample intervals from smallest anchor size to the biggest, the criteria for levels of are shown as the follow:

$$
\begin{cases}
small\_positive & : a_i \in \{A_1\} \\
medium\_positive & : a_i \in \{A_2, A_3\} \\
big\_positive & : a_i \in \{A_4, A_5\}
\end{cases}
\tag{1}
$$

In Equation (1), $a_i$ represents the area of an anchor box, and A1∼A5 denote the present area of the anchor boxes in 5 different levels. The IOU discrimination threshold of small and medium anchor boxes are then decreased to 0.5 and 0.6 respectively, to ensure more small and medium boxes will be matched. For large anchor boxes, the default discrimination threshold is kept. By lowering the positive sample discrimination threshold, the anchor box is easier to match with the small object area, and the number of positive samples with the small object area is therefore increased.

Theoretically, lowering the matching threshold for the large-size anchor boxes can also effectively increase the number of matching anchor boxes. However, compared with the small object area, the large object has the following two differences:

1. For the large object it is much easier to meet the discrimination condition of the IOU threshold. As it is seen in Figure 3, the large object located of the image has larger number of matching anchor boxes although for a threshold which has not been decreased. This suggests that further reduction of the threshold has only a limited impact on the increase of the positive samples.
2. Since a large object area contains a rich image feature information, compared to small objects, it is easier to obtain a set of valid discrimination and bounding box regression parameters during

the network training process. Therefore, it is very limited to enhance the effect of detection performances for large objects by reducing the IOU discrimination threshold.

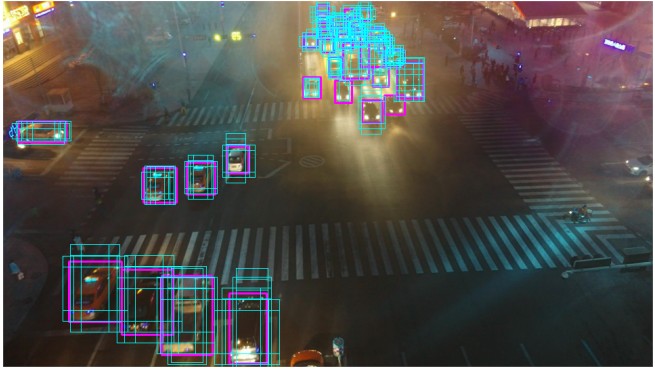

**Figure 3.** Sampling Results Example on Big Objects.

From the network training perspective, the object detection approaches that are limited by computing resources often need to set an upper limit of samples. Part of the sampling results will be discarded randomly when too much samples are collected. Taking the FPN as an example, the upper limit on the total number of samples is usually 256, and arrange for positive and negative samples are 128 respectively, the redundant samples will be discarded. In this paper, we argue that the sample priority of the small object area should be higher than that of the large objects. In cases where there are a combination of small and large objects in the image, first and most important is to ensure a sufficient number of the small object areas samples for section. Therefore, the same IOU threshold value as the original FPN method is maintained for the large object areas, and the number of positive samples is not increased.

For negative sample sampling, besides considering the match of the anchor box and the object size, it is also necessary to consider the effect of different discrimination difficulty on the accuracy of the algorithm. For the object detection algorithm, the proposed approach divide the negative samples into easy and hard negative samples depending on the IOU threshold. In particular, the easy negative samples help the network to converge quickly. The detection accuracy however is mainly dependent on hard negative samples. Therefore, when collecting negative samples, the ratio of the number of hard to the number of easy negative samples is balanced. Figure 4 shows the example result of negative samples, where blue, green and red rectangles denote small, medium and big negative samples, most of them belong are easy and small samples.

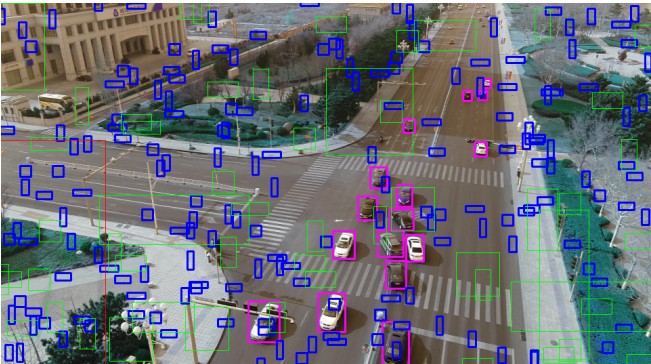

**Figure 4.** Negative Sample Results of Random Sampling.

To address problem above, the Libra RCNN propose a balance sampling method to ensure the diversity of negative samples: First, according to IOU between anchor boxes and ground-truth the divide negative samples into different intervals. Second, divide the number of negative samples equally

according to the intervals and balance sampling in each interval. Within the FPN method, Negative samples are defined as the anchors whose IOU with ground-truth are lower than 0.3, the Libra RCNN further divided it into easy, medium and hard negative intervals which are defined as the follow:

$$
\begin{cases}
easy\_negative : IOU \in [0, 0.1) \\
medium\_negative : IOU \in [0.1, 0.2) \\
hard\_negative : IOU \in [0.2, 0.3)
\end{cases}
\tag{2}
$$

Based on Libra RCNN, a balance negative sampling method which combining samples' scale and difficulty is proposed. Negative samples were divided into 8 intervals as shown in the Equation3. For medium and big negative samples this approach adopt the similar difficulty dividing approach as Libra RCNN, for instance, the easy_negative_medium negative sample denote the samples whose $IOU \in [0, 0.1)$ and scale $a_i \in \{A_2, A_3\}$. For small negative samples, since the IOU discrimination threshold of positive samples are adjusted to 0.5, the dividing approach of Libra RCNN is easy to cause confusing between positive and negative samples, therefore this approach correspond adjusted the dividing approach that only divide them into two different intervals.

$$
\begin{cases}
easy\_negative\_small : IOU \in [0, 0.1), a_i \in \{A_1\} \\
hard\_negative\_small : IOU \in [0.1, 0.2), a_i \in \{A_1\} \\
easy\_negative\_medium : IOU \in [0, 0.1), a_i \in \{A_2, A_3\} \\
medium\_negative\_medium : IOU \in [0.1, 0.2), a_i \in \{A_2, A_3\} \\
hard\_negative\_medium : IOU \in [0.2, 0.3), a_i \in \{A_2, A_3\} \\
easy\_negative\_big : IOU \in [0, 0.1), a_i \in \{A_4, A_5\} \\
medium\_negative\_big : IOU \in [0.1, 0.2), a_i \in \{A_4, A_5\} \\
hard\_negative\_big : IOU \in [0.2, 0.3), a_i \in \{A_4, A_5\}
\end{cases}
\tag{3}
$$

### 3.3. Balanced Sampling

According to the dividing method mentioned above, a balance sampling approach is proposed: the positive samples are balanced according to the scale size to form a positive sample set. For the negative samples determined by the anchor box, the negative sample set is formed by balanced sampling with comprehensive consideration of the difficulty and scale size. For the sample set with an upper limit of N, the sample collection method designed in this paper is demonstrated in Algorithm 1.

---

**Algorithm 1** Balanced Sample Algorithm.

---

**Inputs:**
   Positive/Negative Sample Sets;
 2: Number of Select Samples N;
**Outputs:**
   Sample Set U;
 4: $divide\_num = \frac{N}{set\_num}$
   U = []
 6: sort(Sets)
   for set in Sets:
 8:     if $n_{set} > divide\_num$:
          U.append(sample($n_{set}, divide\_num$))
10:     else:
          U.append($n_{set}$)
12:        reshape($divide\_num$)
   **return** U

---

Ideally, the total number of positive and negative samples should be equal, therefore this approach initializes $divide\_num$ to the average of the total number of sample sets $\frac{N}{set\_num}$, if number of samples of all the intervals satisfies $n_{set} > divide\_num$, it is only needed to randomly sampling in each interval

to generate set $U$. However, the ideal condition mentioned above is hardly appear in actual situation, therefore balance sampling is a problem that should be considered. If the total number of samples is less than the upper sampling limit N, it is necessary to include all samples in the sample set; Otherwise, the number of uniformly sampled objects in each interval, *divide_num*, is calculated based on the interval data, *set_num*. Sampling is then carried out from low to high according to the sample data in each interval. If the number of samples in the current interval, $n_{set} > divide\_num$, then *divide_num* samples are randomly selected to be included in the sample collection of the current interval; otherwise, all *n_set* samples are included in the sample collection, and *divide_num* is adjusted for subsequent sampling intervals using the reshape method.

The key point of the balanced sampling method is the reshape method for $n_{set} < divide\_num$. All samples in these kind of interval should be retained since the demand number of samples if more than the actual collected samples. Since the order of sampling approaching is depend on number of samples in each interval, therefore all of the subsequent intervals are redundant, which means the subsequent intervals are satisfy the following condition:

$$\sum_{i=j+1}^{set\_num} > (set\_num - j) * divide\_num + (divide\_num - num_{set}) \tag{4}$$

In Equation (4), $j$ represents the index of the current interval set in all sorted intervals. Since the surplus samples can be collected in the subsequent sampling process, a sufficient number of samples can be still collected. Therefore, as many as possible samples should be collected from the remaining intervals while maintaining the balance. The reshape method for updating *divide_num* is designed as the follow:

$$num\_divide = \frac{(N - \sum_{i=1}^{j} n_i)}{set\_num_{left}} \tag{5}$$

In Equation (5), $set\_num_{left}$ represents the number of remaining intervals. Since samples of each subsequent interval is updated. Take the collection process of the positive samples as an example and suppose that the number of samples in small_positive intervals, nsmall, is the lowest and less than divide_num. Then, *divide_num* is updated to $(N - nsmall)/2$ for the sampling process in the subsequent intervals. If the number of samples in the medium_positive and big_positive intervals is greater than the updated value of divide_num, then they are uniformly sampled.hrough the balanced sampling method, factors such as scale and difficulty are fully considered in the process of generating the sample set, which can effectively increase the number of small object samples and ensure sample diversity.

*3.4. Loss Function*

Similar to other tow-stage methods, the loss function is defined as sum of classification and regression loss:

$$L_{total} = L_{cls}^{RPN} + L_{bbox}^{RPN} + L_{cls}^{Cat} + L_{bbox}^{Reg} \tag{6}$$

In Equation (6), $L_{cls}$ and $L_{bbox}$ denote classification and regression loss of MB-RPN and finetune loss respectively. Cross entropy is adopted for measuring the classification loss:

$$L_{cls}(y_i, y_i^*) = -[y_i^* log(y_i) + (1 - y_i^*) log(1 - y_i)] \tag{7}$$

where $y_i$ and $y_i^*$ denote the predict and annotated category respectively where $y_i^*$ is 1 if the anchor is positive in MB-RPN and $y_i^*$ is 1 at the dimension representing the object's category in label vector. $L_{reg}$ denote the smooth L1 regression loss [7]:

$$L(t_i, t_i^*) = \begin{cases} 0.5 (t_i - t_i^*)^2, & |x| < 1 \\ |x| - 0.5, & others \end{cases} \tag{8}$$

where $t_i$ and $t_i^*$ denote predict and annotated coordinate and scale transform:

$$
\begin{cases}
t_x = (x - x_a)/w_a & t_y = (y - y_a)/h_a \\
t_w = log(w/w_a) & t_h = log(h/h_a) \\
t_x^* = (x^* - x_a)/w_a & t_y = (y^* - y_a)/h_a \\
t_w^* = log(w^*/w_a) & t_h^* = log(h^*/h_a)
\end{cases}
\tag{9}
$$

*3.5. Discussion*

In this section, both the framework and detail of proposed approach are introduced, the overall architecture is similar to FPN except positive/negative candidate sampling method are adjutsted. First, the framework is introduced, including network architecture, pre-trained backbone, object detection pipeline and network detail. Second, this section represents the matching mechanism multi-scale objects, the IOU threshold for small and medium anchors are reduced to ensure more small objects will be matching successfully. Third, a sampling algorithm is introduced to ensure the diversity of sampling results, all the samples are divided into different intervals, the algorithm tries to sample balance amount of samples in each of the interval. Finally, loss function of the proposed approach is introduced, including the cross entropy loss for classification and smooth L1 loss for localization.

## 4. Experiments

*4.1. Benchmarks*

The proposed approach are evaluated on two datasets: Object Detection in Aerial Images(DOTA) and e Unmanned Aerial Vehicle Benchmark(UAVB) [15,16], the detail of them are as the follow:

1.  DOTA contains over 2000 remote images. All of the images are large size about over 4000 × 4000 pixels. Images are annotated by experts in aerial and remote image interpretation using 15 common object categories, such as plane, ship, harbor, etc. The objects' distribution of each category are shown in Figure 5a, the abbreviation of each category will be shown in Table 2.
2.  UAVB contains a unmanned aerial vehicle dataset, each frame is of the size about 560 × 1000 pixels and contains high density small objects. Vehicle category include car, truck and bus. The objects' distribution of each category are shown in Figure 5b.

**Table 2.** Quantitative performance(AP%) of our model on DOTA benchmark datasets compared with comparison approaches. The best performance on each category is colored in red.

|  | SSD | RetinaNet | SRetinaNet | FPN | GA-RPN | Libra RCNN | MB-RPN |
|---|---|---|---|---|---|---|---|
| plane (pl) | 81.3 | 88.7 | 86.2 | 88.9 | 88.4 | 89.7 | 90.2 |
| baseball-diamond (bd) | 50.6 | 64.5 | 57.8 | 75.8 | 75.2 | 73.6 | 77.0 |
| bridge (bg) | 39.1 | 47.5 | 21.4 | 48.6 | 43.1 | 49.7 | 53.0 |
| ground-track-field (gtf) | 44.2 | 49.0 | 49.5 | 53.9 | 53.5 | 54.8 | 53.3 |
| small-vechicle (sv) | 56.0 | 58.1 | 61.1 | 63.8 | 67.7 | 70.3 | 70.4 |
| large-vehicle (lv) | 54.2 | 57.8 | 59.0 | 63.6 | 66.3 | 65.0 | 66.0 |
| ship (sp) | 61.1 | 69.2 | 74.4 | 76.7 | 77.0 | 77.1 | 76.8 |
| tennis-court (tc) | 84.3 | 88.2 | 70.3 | 90.7 | 90.1 | 90.8 | 90.8 |
| basketball-court (bc) | 68.6 | 73.5 | 58.1 | 78.2 | 78.4 | 75.0 | 78.9 |
| storage-tank (st) | 61.5 | 75.6 | 80.5 | 81.0 | 84.1 | 83.0 | 83.7 |
| soccer-ball-field (sbf) | 21.3 | 28.4 | 16.7 | 36.5 | 36.8 | 37.2 | 41.2 |
| roundabout (ra) | 43.6 | 49.1 | 55.2 | 56.8 | 58.9 | 60.0 | 59.4 |
| harbor (hb) | 51.6 | 56.7 | 35.9 | 67.3 | 66.7 | 67.6 | 68.3 |
| swimming-pool (st) | 47.4 | 61.1 | 64.3 | 71.1 | 72.0 | 72.6 | 70.9 |
| helicopter (hl) | 28.0 | 38.6 | 57.1 | 44.7 | 50.3 | 62.6 | 56.5 |
| mAP | 56.9 | 60.6 | 58.1 | 65.5 | 66.8 | 67.6 | 68.5 |

Both the DOTA and UAVB dataset contain all kinds scale and small objects account a large proportion. It means that both the ability of detecting small objects and all the scale of objects are important.

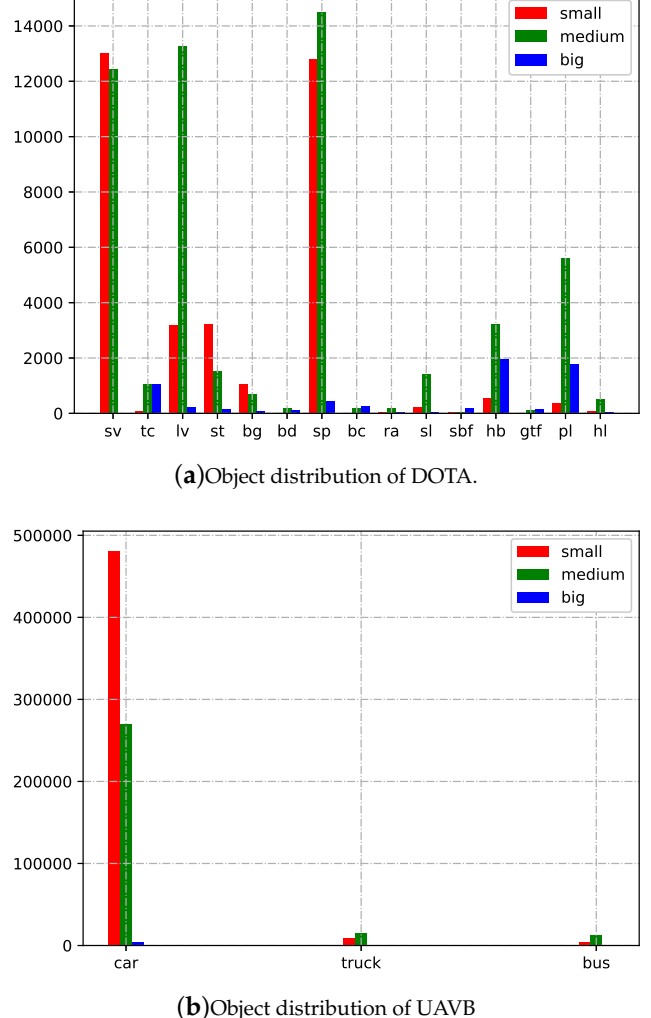

(**a**)Object distribution of DOTA.

(**b**)Object distribution of UAVB

**Figure 5.** Objects distribution of DOTA and UAVB dataset.

### 4.2. Implementation Detail

The network is established on Tensorflow and trained end-to-end.MB-RPN loss and detection loss are optimized simultaneously with Nvidia 1080Ti on Ubuntu operation system [17]. ResNet-101 pre-trained network is adopted to extract image features and other convolution layers were initialized randomly [14]. To optimize network parameters, Adam optimizer with lr=$10^{-6}$, $\beta_1 = 0.9$ and $\beta_2 = 0.999$ is adopted [18].

The input images were set to $600 \times 600$. For DOTA dataset, since the shape of training and testing images are much larger than input size, to reduce the loss of image resolution the images are cropped into input size with stride of 300 for training and testing and merge test results to original shape. For UAVB dataset, since the shape of training and testing images are similar to input size, it is only needed to resize the images to input size. The size of anchor boxes for layer Conv1∼Conv5 are [32,64,128,256,512], which is consistent to the default value of FPN method. For DOTA dataset, aspect ratios of anchor boxes is [$\frac{1}{7}$,$\frac{1}{5}$,$\frac{1}{3}$,$\frac{1}{2}$,1,2,3,5,7] to adapt categories with both normal and slender shape such as bridge. For UAVB dataset, since the shape of all the categories are normal, therefore the

aspect ratio is same to default FPN method. Mean average precision(mAP) is adopted to evaluate the proposed approach[19].

### 4.3. Effectiveness of Multi-Scale Sampling on Positive Sample

To evaluate the effectiveness of Multi-scale sampling for positive samples, the comparison of positive samples distribution on different scales between multi-scale and origin FPN sampling method on DOTA dataset, the result are shown in Figure 6. Since the IOU discrimination thresholds of small and medium objects are reduced, the amount positive samples are highly improved, which means more small anchor boxes are matched. The matching results are also visualized, compare to Figure 7 the matching results has been effectively improved.

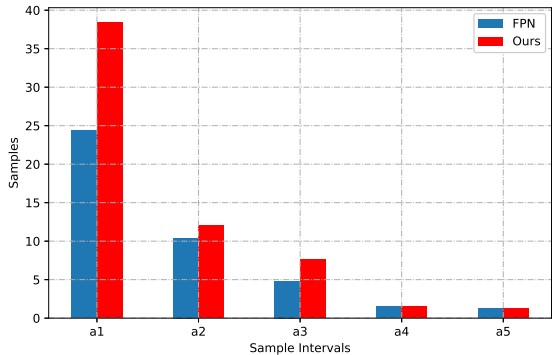

**Figure 6.** Distribution of multi-scale sampling and origin FPN sampling method.

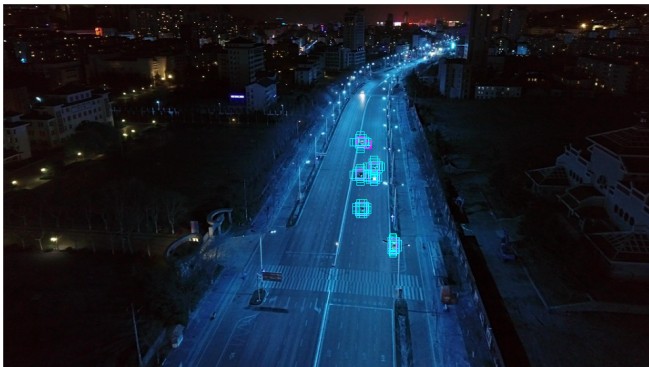

**Figure 7.** Example of Multi-scale Sampling Results.

### 4.4. Effectiveness of Multi-Scale Balanced Sampling on Negative Samples

To evaluate the effectiveness of multi-scale balanced sampling, comparison of negative samples' distribution on different scales and IOU between MB-RPN, FPN and Libra RCNN sampling method on DOTA dataset, the result are shown in Figure 8, where $b_1 \sim b_8$ denote total amount of hard_small to easy_big negative samples. Most of the samples are Easy_small in FPN method, the Libra RCNN alleviate this situation significantly but the majority is still easy samples($b_1$, $b_3$ and $b_6$), especially the easy_small samples, the multi-scale sampling further improved samples distribution situation. The matching results are also visualized in Figure 9, compare to FPN and Libra RCNN, the matching results has been effectively improved.

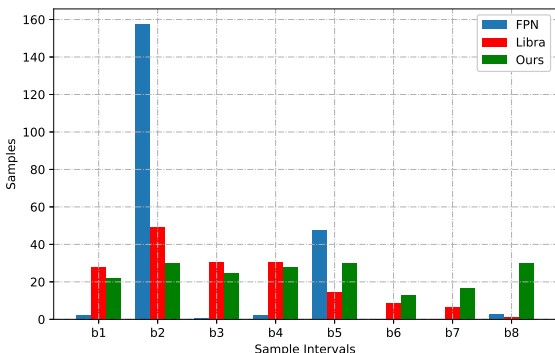

**Figure 8.** Distribution of multi-scale balanced,FPN and Libra RCNN sampling method.

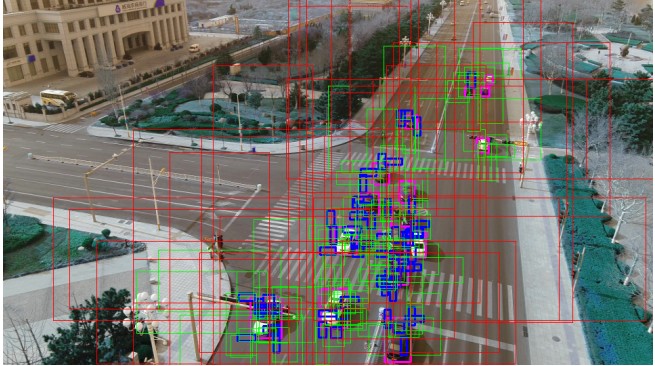

**Figure 9.** Example of Multi-scale Balanced Sampling Results of FPN, Libra RCNN and Multi-scale Balanced.

## 4.5. Performance Comparison with Other Method

Several one stage and two stage object detection methods is adopt to evaluate the effectiveness of MB-RPN: SSD [3] RetinaNet [10], FPN [8], Libra RCNN [11], GA-RPN [13] and SRetinaNet [12]. All of these methods except SRetinaNet are implemented with the source code provided by authors. For SRetinaNet method, it is implemented by adjusting hyperparameters of RetinaNet.

Table 2 shows the quantitative results on DOTA, the best performance for each category is colored in red. The mAP of MB-RPN achieves 68.5%, which outperform other methods. Compare with the one-stage approaches, since they are lack of positive/negative discrimination process, the detection accuracy are lower than all of the two-stage approaches obviously. Compare with original FPN, the mAP is 3% higher, moreover the AP of each category is also higher. Compare with GA-RPN and Libra RCNN, the mAP is increased about 1.7and 0.9% respectively, the AP of the most of categories are increased. The visualization comparison between MB-RPN and Libra RCNN is shown in Figure 10, MB-RPN detects more accurate small objects in various challenging cases, e.g., small vehicle objects at bottom left of Figure 10a and middle of Figure 10b. At the same time, the performances of other medium and big objects are not decreased. The above phenomenon proves the effectiveness of MB-RPN in enhancing the detection performances for images with small objects and large scale variants.

Considering the performances gap between one-stage and two-stage approaches, in this paper the performance comparison of UAVB dataset is only carried out between FPN, GA-RPN, Libra RCNN and MB-RPN. Table 3 shows the quantitative evaluation of these approaches, the best performance for each category is colored in red. All of the mAP acquired from the approaches above are not ideal, this may because the imbalance of samples. However, the MB-RPN still largely outperform other approaches on both mAP and AP for each category. Figure 11 provides a visual comparison of our approach and Libra RCNN, since both of the two approaches' performance are not ideal, it only shows

the localization results but without categories. It can be seen that MB-RPN detects more small objects such as cars at upper of the input image which is corresponding to sampling results and distribution shown above. The above phenomenon proves the effectiveness of MB-RPN in enhancing the detection performances.

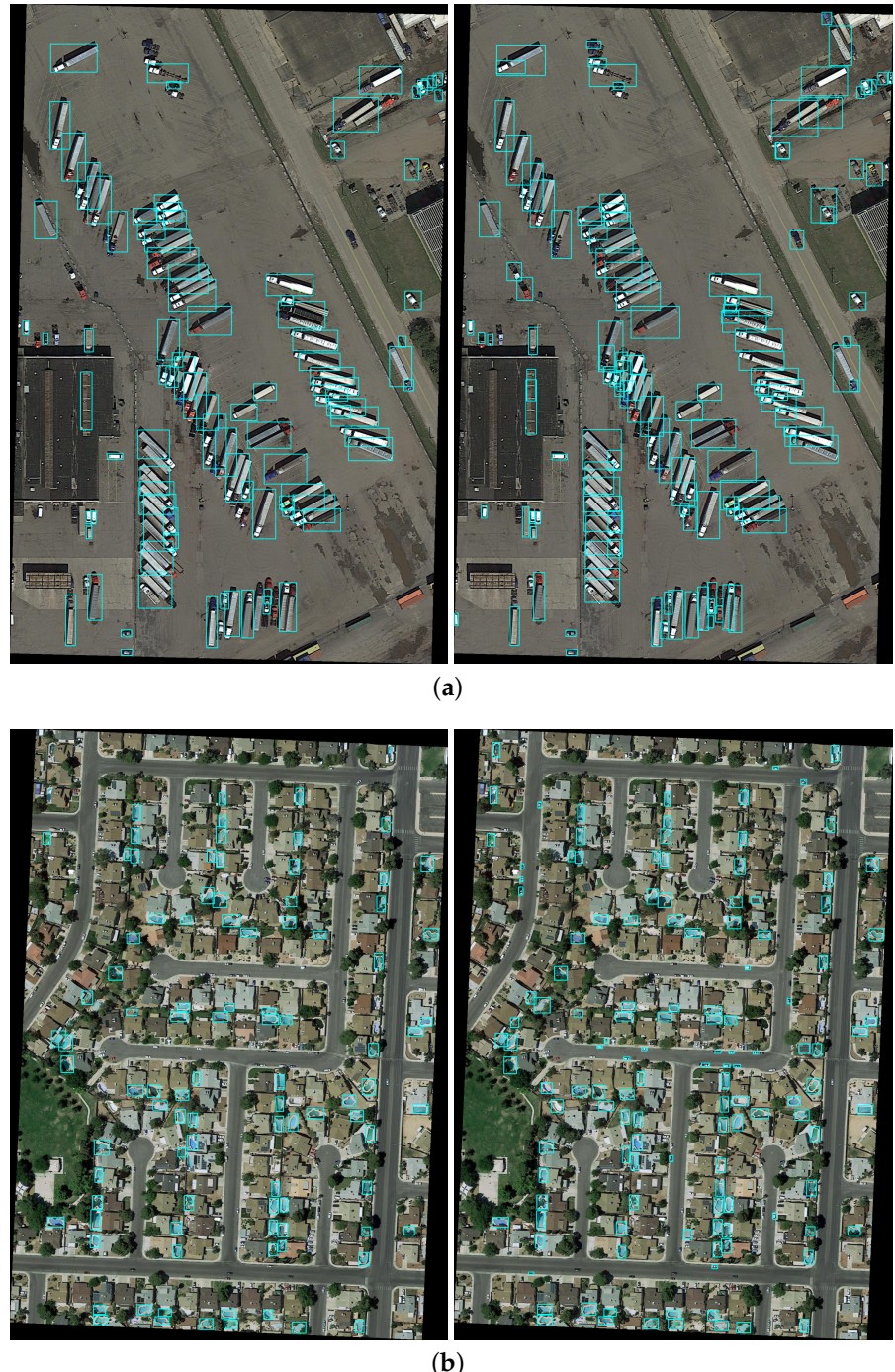

**Figure 10.** Selected visual comparison of DOTA benchmark dataset. Each subfigure include result of Libra RCNN on the left and MB-RPN on the right. (**a**) small objects (**b**) middle objects.

**Table 3.** Quantitative performance(AP%) of our model on UAVB benchmark datasets compared with comparison approaches. The best performance on each category is colored in red.

|       | FPN  | GA-RPN | Libra RCNN | MB-RPN |
|-------|------|--------|------------|--------|
| car   | 45.1 | 47.2   | 48.8       | 51.2   |
| bus   | 5.3  | 8.6    | 7.2        | 21.7   |
| truck | 28.2 | 31.5   | 31.1       | 33.2   |
| mAP   | 21.1 | 21.7   | 23.0       | 29.4   |

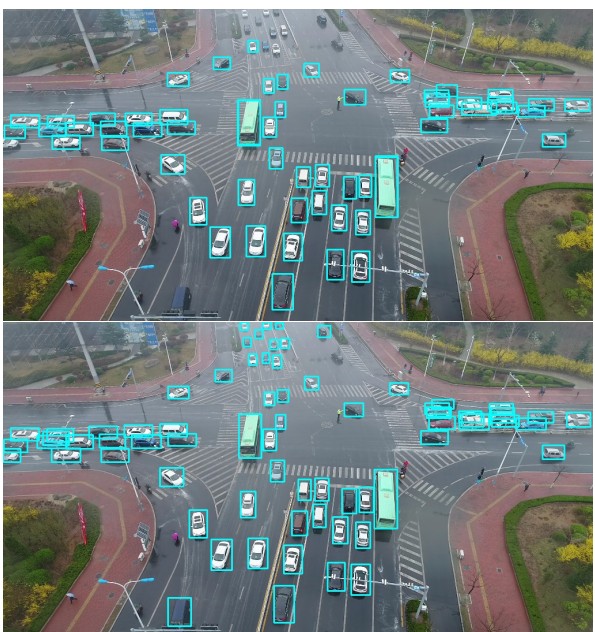

**Figure 11.** Selected visual comparison of DOTA benchmark dataset which includes Libra RCNN on the top and MB-RPN on the bottom.

### 4.6. Disscusion

In this section, the experimental settings and results are introduced. First, the adopted DOTA and UAVB dataset are introduced, including their amounts of training/testing images and the distribution of each category samples. Second, implement details are represented, including the input size, crop mechanism of large images, parameter initialization and optimization method and hardware/software platform. Finally, both quantitative and visualized comparison are represented, the results show that under the equal conditions MB-RPN outperform other similar methods.

### 5. Conclusions

In this paper, a multi-scale balanced sampling approach for detecting small objects in complex scenes is proposed. With multi-scale positive sampling method, more small objects is able to be included in the network training process. With the balanced negative sampling method, the diversity of negative samples is ensured. Experimental results shows that compare with other similar methods, this approach acquire better performances on the images with small objects and large scale variant objects.

**Author Contributions:** Conceptualization, H.Y.; methodology, H.Y., J.G.; software, H.Y.; validation, H.Y.; writing—original draft preparation, H.Y.; writing—review and editing, D.C.; supervision, D.C.; funding acquisition, D.C. and J.G. All authors have read and agreed to the published version of the manuscript.

**Funding:** This research was funded by Beijing Municipal Natural Science Foundation (4182031).

**Conflicts of Interest:** The authors declare no conflict of interest.

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
