# Peer review of "Object Detection Using Multi-Scale Balanced Sampling"

_applsci, doi:10.3390/app10176053_

Round 1

Reviewer 1 Report

The paper has 26% plagiarism  with word count 10. This much plagiarism is not acceptable for any reputed journal.

Author Response

Point 1: The paper has 26% plagiarism  with word count 10. This much plagiarism is not acceptable for any reputed journal.

Response 1: We have re-organized the language to avoid plagiarism of our paper. However, we cannot avoid the plagiarism of references. We hope the revised version could meet the requirements.

Reviewer 2 Report

This paper focuses on the detection of small objects in images. The authors address the problem by adjusting samples’ matching conditions to match with objects’ scale differences so that more samples are included during the training process. Furthermore, they divide the sample set into multiple intervals according to scale and corresponding difficulties to preserve the diversity of samples.

It is not clear whether the proposed approach is end-to-end trainable or not. This work’s main focus is adjusting/optimizing the anchor boxes to include a large and balanced number of samples from all classes. However, there are existing techniques that can be used for anchor optimization and most of them are end-to-end trainable. Please include comparison results and properly cite the recently published related papers and explain how your approach is better than previous work. Some of the papers related to this work that need to be considered for a complete literature review:

Zhong Y, Wang J, Peng J, Zhang L. Anchor box optimization for object detection. InThe IEEE Winter Conference on Applications of Computer Vision 2020 (pp. 1286-1294).

Yang T, Zhang X, Li Z, Zhang W, Sun J. Metaanchor: Learning to detect objects with customized anchors. InAdvances in Neural Information Processing Systems 2018 (pp. 320-330).

Wang J, Chen K, Yang S, Loy CC, Lin D. Region proposal by guided anchoring. InProceedings of the IEEE Conference on Computer Vision and Pattern Recognition 2019 (pp. 2965-2974).

Ahmad M, Abdullah M, Han D. Small Object Detection in Aerial Imagery using RetinaNet with Anchor Optimization. In 2020 International Conference on Electronics, Information, and Communication (ICEIC) 2020 Jan 19 (pp. 1-3). IEEE.

In section 3.3. Balanced Sampling, authors propose a balanced sampling method based on the scale and difficulty level. However, it is not clearly explained the exact methodology used to measure the difficulty of a sample. 

In Figure 1, “MB-RPN Classification” has two heads, please explain why there are two blocks of the same module, similarly at the end of architecture “Detection Classification” is also drawn twice. Please explain. 

Authors provide comparisons with straightforward backbones such as SSD, RetinaNet, FPN etc. However, these approaches are very old. In order to validate the results a more recent comparative study shall be conducted. 

Author Response

Point 1: It is not clear whether the proposed approach is end-to-end trainable or not. This work’s main focus is adjusting/optimizing the anchor boxes to include a large and balanced number of samples from all classes. However, there are existing techniques that can be used for anchor optimization and most of them are end-to-end trainable. Please include comparison results and properly cite the recently published related papers and explain how your approach is better than previous work. Some of the papers related to this work that need to be considered for a complete literature review:

Zhong Y, Wang J, Peng J, Zhang L. Anchor box optimization for object detection. InThe IEEE Winter Conference on Applications of Computer Vision 2020 (pp. 1286-1294).

Yang T, Zhang X, Li Z, Zhang W, Sun J. Metaanchor: Learning to detect objects with customized anchors. InAdvances in Neural Information Processing Systems 2018 (pp. 320-330).

Wang J, Chen K, Yang S, Loy CC, Lin D. Region proposal by guided anchoring. InProceedings of the IEEE Conference on Computer Vision and Pattern Recognition 2019 (pp. 2965-2974).

Ahmad M, Abdullah M, Han D. Small Object Detection in Aerial Imagery using RetinaNet with Anchor Optimization. In 2020 International Conference on Electronics, Information, and Communication (ICEIC) 2020 Jan 19 (pp. 1-3). IEEE.

Response 1:

In section 1 we made a statement that our approach is end-to-end:To address the problems above, we propose an end-to-end object detection approach with multi-scale balanced sampling. In summary, the key contributions of our work are as follows.Also, in section  4.2: We train our model end-to-end, MB-RPN loss and detection loss are optimized simultaneous.

In section 2 we introduced two of the papers: ‘Region proposal by guided anchoring’ and ‘Small Object Detection in Aerial Imagery using RetinaNet with Anchor Optimization’.

Point 2: In section 3.3. Balanced Sampling, authors propose a balanced sampling method based on the scale and difficulty level. However, it is not clearly explained the exact methodology used to measure the difficulty of a sample. 

Response 2: We add the introduction of difficulty measurement method and the dividing approach: the definition of difficulty is the same as Libra RCNN, and the our balance sampling approach consider both scale and difficulty.

Point 3: In Figure 1, “MB-RPN Classification” has two heads, please explain why there are two blocks of the same module, similarly at the end of architecture “Detection Classification” is also drawn twice. Please explain. 

Response 3: This is a mistake, similar to faster rcnn, the two blocks should be classification and regression respectively, we have revised it.

Point 4: Authors provide comparisons with straightforward backbones such as SSD, RetinaNet, FPN etc. However, these approaches are very old. In order to validate the results a more recent comparative study shall be conducted. 

Response 4: Since we added the introduction of ‘Region proposal by guided anchoring’ and ‘Small Object Detection in Aerial Imagery using RetinaNet with Anchor Optimization’ in related work. We also add the evaluation results of them on Section 4. The source code of guided anchoring are provided by author on the open source lib of mmdetection. For another paper, we adjust the parameter on RetinaNet to meet the author’s setting.

Reviewer 3 Report

1. The introduction needs to be broadened and deepened regarding the proposal of this article.

2. The dataset characteristics are unclear, how many images of bus, car, truck, and etc. are not explained. In addition, whether the class in the dataset is an imbalance or not?

3. Instead of listing several pre-trained networks and their accuracy, it is better to also explain what is the objective/advantage of detecting small objects in the abstract section?. This will allow the reader to easily grab the big picture of the proposed system.

Author Response

Point 1: The introduction needs to be broadened and deepened regarding the proposal of this article.

Response 1:

We revised the introduction and related works, more related references are added. Besides, we add the importance of object detection algorithms in images with large scale variants which is one of the effectiveness of multi-scale balance sampling.

Point 1: The dataset characteristics are unclear, how many images of bus, car, truck, and etc. are not explained. In addition, whether the class in the dataset is an imbalance or not?
Response 2:

We add distribution of each category of DOTA and UAVB in section 4.

Point 3: Instead of listing several pre-trained networks and their accuracy, it is better to also explain what is the objective/advantage of detecting small objects in the abstract section?. This will allow the reader to easily grab the big picture of the proposed system.
Response 3:

We narrated the enhancement of detection performances compare with similar approach(Libra RCNN) in abstract.

Round 2

Reviewer 1 Report

1. Third form of sentence must be used and please try to avoid words like I, We etc.

2. Abstract should be rewritten in more effective manners. 

3. Introduction is not proper and it must contains some more details of the system in which authors want to work.

4. Discussion about related work is too less and it must be some appropriate and more literature survey is requires as a lot of work already done in the same field.

5. Description of the system in which all the algorithms and proposed methods are implemented is missing.

6. Unit of measurement is missing. The results are quantified but what is the unit and on what basis it is compared is not found anywhere in the paper.

7. A lot of abbreviations are used without giving their full form. First time use of any abbreviation must be along with its full form like DOTA, UAVB etc.

8. More discussion in results are required. Authors wind up things very early without proper discussion.

9. while removing plagiarism, in some places, the language is very confusing.

10. Rewrite the paper with proper discussion in each section, proper explanation of of things and methods along with description of system and units.

11. Unit on the basis of which comparison is performed must be clearly explained.

Author Response

Point 1: Third form of sentence must be used and please try to avoid words like I, We etc.

Response 1: We have examined the paper and tried to avoid We, Our.

Point 2: Abstract should be rewritten in more effective manners. 

Response 2: We rewrite the abstract: add the issue that we want to solve.

Point 3: Introduction is not proper and it must contains some more details of the system in which authors want to work.

Response 3: In the introduction section, we further refine the development of object detection, the problem need to be solved in dealing with small objects and various scale objects.

Point 4: Discussion about related work is too less and it must be some appropriate and more literature survey is requires as a lot of work already done in the same field.

Response 4: We added the process of RCNN-Fast RCNN-Faster RCNN, introduced the process from pipeline deep neural network based approach to end-to-end algorithms.

Point 5: Description of the system in which all the algorithms and proposed methods are implemented is missing.

Response 5: We rewrite the implement detail of our approach in section 4, and add the implement method of comparison approaches.

Point 6: Unit of measurement is missing. The results are quantified but what is the unit and on what basis it is compared is not found anywhere in the paper.

Response 6: We change the unit of quantitative results in percentage,  in  the implement detail we had introduced the performances evaluation method(mAP as PASCAL Dataset)

Point 7: A lot of abbreviations are used without giving their full form. First time use of any abbreviation must be along with its full form like DOTA, UAVB etc.

Response 7: All the abbreviations, at the first time we add its full form, like DOTA UAVB IOU ROI etc.

Point 8: More discussion in results are required. Authors wind up things very early without proper discussion.

Response 8: Based on current discussions, we add the comparison between our and other three approaches on DOTA. Besides, we discussed the reason why all performances of UAVB are not ideal and show the enhancement of our method.

Point 9: while removing plagiarism, in some places, the language is very confusing.

Response 9: We have examined the paper and tried to avoid this kind of confusing.

Point 10: Rewrite the paper with proper discussion in each section, proper explanation of of things and methods along with description of system and units.
Response 10: We add discussion on section 3 and 4, summarized the main work.

Point 11: Unit on the basis of which comparison is performed must be clearly explained.

Response 11: Because we changed the unit of mAP to percentage, therefore this problem has been solved.

Reviewer 2 Report

The authors have included recent published articles in literature review and comparison results are also updated accordingly. 

The difficulty criteria for samples is included which was missing previously.

Figure 1 is updated and seems satisfactory.

Comparison results are updated with recent works.

In the light of above comments, I am satisfied with the current version of the paper as the authors have responded to all the raised concerns in a satisfactory manner. 

Author Response

Dear reviewer,

    Thank you for your recognition.